# A Systematic Review and Meta-Analysis of the Impact of Mindfulness Based Interventions on Heart Rate Variability and Inflammatory Markers

**DOI:** 10.3390/jcm8101638

**Published:** 2019-10-07

**Authors:** Lina Rådmark, Anna Sidorchuk, Walter Osika, Maria Niemi

**Affiliations:** 1Department of Clinical Neuroscience, Karolinska Institutet, 171 77 Stockholm, Sweden; 2Center for Social Sustainability, Department of Neurobiology, Care Sciences and Society, Karolinska Institutet, 171 77 Stockholm, Sweden; 3Centre for Psychiatry Research, Department of Clinical Neuroscience, Karolinska Institutet, 171 77 Stockholm, Sweden; 4Stockholm Health Care Services, Region Stockholm, CAP Research Centre, 113 30 Stockholm, Sweden; 5Northern Stockholm Psychiatry, Stockholm Health Care Services, Region Stockholm, 112 81 Stockholm, Sweden; 6Department of Public Health Science, Karolinska Institutet, 171 77 Stockholm, Sweden

**Keywords:** Mindfulness Based Interventions, Interleukins, C-reactive protein, heart rate variability, meta-analysis

## Abstract

Mindfulness Based Interventions (MBIs) have recently been increasingly used in clinical settings, and research regarding their effects on health has grown rapidly. However, with regard to the physiological effects of mindfulness practices, studies have reported associations that vary in strength and direction. Therefore, in this systematic review and meta-analysis, we aimed to systematically identify, appraise, and summarize the existing data from randomized and non-randomized controlled trials that examine physiological effects of the standardized MBIs by focusing on pro-inflammatory cytokines and C-reactive protein, and commonly used heart rate variability parameters. The following electronic databases were searched: MEDLINE (via Ovid), PsychINFO (via Ovid), PubMed, Web of Science, EMBASE, CINAHL, ProQuest (Dissertations and Theses), and Centre for Reviews and Dissemination. The systematic review identified 10 studies to be included in the meta-analysis, comprising in total 607 participants. The meta-analysis ended up with mixed and inconclusive results. This was assumedly due to the small number of the original studies and, in particular, to the lack of large, rigorously conducted RCTs. Therefore, the current meta-analysis highlights the necessity of larger, more rigorously conducted RCTs on physiological outcomes with standardized MBIs being compared to various forms of active controls, and with more long-term follow-ups.

## 1. Introduction

In recent decades, the use of Mindfulness Based Interventions (MBIs) in clinical settings, and research regarding their effects on various measures has increased rapidly. To date, Mindfulness Based Stress Reduction (MBSR) and Mindfulness Based Cognitive Therapy (MBCT) are the most commonly used MBI programs offered in clinical settings and have the most research support [1]. A recent meta-analysis including altogether 142 non-overlapping samples and 12,005 individuals where MBIs were compared to either no treatment or various forms of active treatment showed the most consistent evidence in support of efficacy of mindfulness for treatment of depression, pain conditions, smoking, and addictive disorders [1]. Though MBIs are not intended to replace standard psychiatric care [2], they show promising results as evidence-based treatment [1]. 

The mechanisms underlying the benefits of MBIs are suggested to include improved emotional regulation strategies and self-compassion levels, decreased rumination and experiential avoidance [3], as well as improved meta-cognitive skills and body awareness [4,5]. A number of authors have suggested models to explain the psychological mechanisms by which mindfulness interventions have an effect [6,7,8], and Hötzel et al. [9] have proposed a theoretical framework that integrates earlier models. This framework proposes that there are four main mechanisms: (1) attention regulation; (2) body awareness; (3) emotion regulation; and (4) change in perspective of the self; these, therefore, together improve self-regulation [9]. 

With regard to physiological effects, a recent case study has given preliminary indication that mindfulness practice may increase heart rate variability (HRV) [10]. HRV refers to the beat-to-beat variation of the R wave to R wave interval between adjacent depolarization of the heart (mirrored by the QRS complexes, i.e., the complex of the Q-Wave, R-wave and S-wave on electrocardiograph recordings) [11,12]. HRV reflects regulation of autonomic balance, e.g., blood pressure, gas exchange, gut, heart, and vascular tone [13]. HRV has been shown to be reduced in patients with anxiety or depressive disorders [14]. The exact pathophysiology is not yet fully understood, but studies seem to indicate that reduced central parasympathetic activity contributes to the reduction of HRV [14]. Studies have also shown a correlation between high HRV and the individual’s capacity to self-regulate attention, emotions, and behavior [10,15]. These initial findings seem thus to indicate that HRV may be a measurable physiological correlate with the neurological capacity for emotional self-regulation [10,15]. The HRV measurements commonly focus on frequency domain parameters (e.g., low frequency (LF), high frequency (HF), LF/HF ratio), and time domain parameters (e.g., standard deviations of normal-to-normal R-R intervals (SDNN), root mean square standard deviations of R-R intervals (RMSSD)) [16].

Scientific interest in the effects of MBI on the immune system is also growing since accumulating evidence indicates that inflammation may trigger changes that contribute to the pathophysiology of depression and stress-related disorders [17,18,19]. Also, inflammation is one of the aspects of immunity that is regulated by the stress response [20]. Inflammation is a complex process that includes a number of biological markers, many of them classified as cytokines and chemokines, key regulators of immune function with different roles in the inflammatory processes (for example, some of these mediators are predominantly pro-inflammatory, whereas others are mainly anti-inflammatory) [21,22]). Some of the inflammatory markers are considered as to be (potentially) significant for depression, e.g., the pro-inflammatory cytokines as interleukin-6 (IL-6), interleukin-1 (IL-1), and tumor necrosis factor (TNF-α), as well as the acute phase reactant protein C-reactive protein (CRP) [23,24,25].

Overall, studies on physiological effects of mindfulness practices have been increasing in recent years, but the results of individual studies vary in strength and direction of the observed effects. A recent systematic review on mindfulness meditation provides tentative evidence of MBSR and MBSR-derived interventions to modulate some specific immune system biomarkers, although a substantial heterogeneity across individual studies precluded the quantitative synthesis of study effects [26]. Meta-analytical data on inflammatory biomarkers [27] and HRV [28] are available from systematic reviews on the overall effects of mind-body therapies, including Tai Chi, Yoga, Qi Gong, and meditation, while the effects of standardized MBIs on corresponding outcomes remain unclear. Therefore, in this review, we aimed to systematically identify, appraise, and summarize the existing data from randomized and non-randomized control trials that examine physiological effects of standardized MBIs by focusing on inflammatory biomarkers, including pro-inflammatory cytokines and CRP, and on commonly used frequency domain and time domain HRV parameters. 

## 2. Methods

The study was conducted in accordance with the guidelines suggested by the Cochrane handbook for systematic reviews of interventions [29], and the findings and procedure were reported in relation to the Preferred Reporting Items for Systematic Reviews and Meta-Analyses (PRISMA) statement [30]. The protocol was registered with PROSPERO (number 2019 CRD42019136595) and is available online.

### 2.1. Eligibility Criteria

The inclusion and exclusion criteria for the original studies were formulated as the PICOS components (Population, Intervention, Comparator, Outcome, Study design) and consisted of the following: (i) *P*—individuals aged 18 years and above regardless of health status (i.e., studies that recruited patients with somatic or psychiatric disorders were considered for inclusion); (ii) *I*—MBI defined as a 6- to 10-week long intervention with weekly meetings of at least 2 h and home practice assignments, and where formal mindfulness practices constituted a central intervention component. The definition followed the manual of the most evidence-based versions of MBIs [31], but allowing for some variation, since these programs have been adapted in length and content for various populations, as for example, a 6-week MBSR for breast cancer patients [32] or a 9-week Mindfulness-Based Childbirth and Parenting [33] (iii) *C*—any type of active comparators (e.g., a control intervention other than MBI) or inactive comparators (e.g., wait-list or treatment as usual); (iv) *O*—inflammatory biomarkers (including pro-inflammatory cytokines and CRP) and HRV frequency domain and time domain parameters; (v) *S*—randomized controlled trials (RCTs) and non-randomized controlled trials with a matched control group. Studies were excluded if written in a language other than English, or if no numerical data was provided for calculation of the effect size (ES).

### 2.2. Search Strategy 

A literature search strategy was developed in collaboration with two experienced university librarians (CG and KM; see Acknowledgments). The following electronic databases were searched from inception to 9th November 2017: MEDLINE (via Ovid), PsychINFO (via Ovid), PubMed, Web of Science, EMBASE, and CINAHL. Grey literature search was conducted in ProQuest (Dissertations and Theses) and ClinicalTrails.gov. Free text or index terms (e.g., Medical Subject Headings) were searched for in titles, abstracts and key words (see Appendix A for details on searched terms and the use of Boolean operators). The use of key words was adapted to each electronic database (see Appendix A for details). In addition, articles were identified through backward and forward hand-search reference chaining [34]. 

### 2.3. Study Selection and Data Extraction

Literature screening was performed independently by two researchers (M.N. and A.S. or L.R.) who initially assessed the eligibility of the articles by screening the titles and abstracts and, if found relevant, by further examining the full-texts against the eligibility criteria [35]. Selected articles were examined on potential overlap in study populations, which was not found. 

Data extraction was performed independently by two researchers (M.N. and L.R.) and included data on first author’s name, publication year, study design, setting and funding, inclusion and exclusion criteria, sample characteristics (sample size, mean age, health status, ethnicity, and gender), intervention content (type, duration, number and length of sessions), type of comparison, outcome definition and measures (including quantitative data such as means and standard deviations (SDs) or standard error (SE), whichever reported), percent of withdrawals, and study quality (described below). Any disagreements at the stage of screening or data extraction were resolved through discussion and consulting a third reviewer, if necessary. If multiple outcomes were reported in the same study, quantitative data were extracted separately for each outcome. If data were missing in the original reports, authors had been contacted for further clarification.

### 2.4. Quality Assessment

The methodological quality of included studies was assessed independently by two researchers (M.N. and L.R.) using the Cochrane Collaboration’s tool for assessing risk of bias [35], focusing on the following domains: sequence generation and allocation concealment (selection bias), blinding of participants and providers (performance bias), blinding of outcome assessors (detection bias), incomplete outcome data (attrition bias), and selective outcome reporting (reporting bias).

### 2.5. Statistical Analysis

As there were variations in measurements of outcomes, a standardized mean difference using Hedges’ *g* was chosen as a common ES. Because the original studies did not report the changes from baseline, apart from Lee at al [36] and Fogarty et al. [37] who reported both the baseline and post-intervention values and changes from baseline, we focused on post-intervention measurements for consistency, as suggested by the Cochrane handbook for systematic reviews of interventions [29]. Hedges’ *g* were calculated at post-intervention as a difference in means between intervention and control group, divided by the pooled within-group SD and incorporating a correction factor for small sample sizes [38]. Studies that provided no outcome measures at post-intervention were excluded. Throughout quantitative synthesis, the original direction of scales indicating the improvement of outcomes was kept. Thus, for all inflammatory markers and the HRV measure LF and LF/HF, ESs below zero pointed to superiority of the intervention group over the controls, while for the other HRV measures such as HF, SDNN, and RMSSD, ESs above zero indicated that the results favoured the intervention. For interpretation of Hedges’ *g*, we applied Cohen’s convention with the ES defined as small (0.20–0.49), medium (0.50–0.79), and large (≥0.8) [39].

For studies by Creswell et al. [40] and Nyklicek et al. [41] that reported means and SE as the outcome measures, we recalculated SD by multiplying SE by a squared root of the size of the group for which SD is being re-calculated. HRV data from Nyklicek et al. [41] were normalized by the authors for the measures of LF and HF (LF.nu = LF/(total power − very low frequency) and HF.nu = HF/(total power − very low frequency)), enabling comparison of the frequency-domain measurements of two subjects despite wide variation in specific band power and total power. There were no differences in their findings if absolute power instead of normalized units of HF and LF were used. Furthermore, for study by Bower et al. [42] where measures were log-transformed for all outcomes, and for study by Owens et al. [43] with log-transformed results for LF/HF ratio, the corresponding means and SD were converted to raw means and SDs. The converting was made to avoid combining measures on the raw and logarithmic scales together, using the formulas provided by Fu et al. [44] and Higgins et al. [45]. 

We performed meta-analysis separately for each specific outcome originally reported. We used a random-effects model incorporating both within- and between-study variability for quantitative synthesis given the initial assumptions of between-study heterogeneity. Statistical heterogeneity among the studies was evaluated using Q and I^2^ statistics. For Q statistics, *p*-value < 0.1 was regarded as representative of statistically significant heterogeneity, and I^2^ values of 25%, 50%, and 75% indicated low, moderate, and high heterogeneity, respectively [46]. We conducted leave-one-out influence analysis to assess the potential impact of individual studies on the overall pooled ES by omitting one study at a time [47]. For each outcome with three or more studies included, we assessed the presence of publication bias using funnel plots, Egger’s regression asymmetry test [48], and the Begg–Mazumdar adjusted rank correlation test [49]. We have planned to perform a series of subgroup analyses by stratifying the main analysis by a priori identified moderators if at least two studies were included in each subgroup. The small number of studies per outcome precluded such analyses. As prior research indicated, the potential physiological effects of MBIs and other mind-body therapies may vary substantially among individuals with different ill-heath status [26,27], therefore we performed sensitivity analyses for each specific outcome by repeating the main analysis after excluding studies where study population was consisted of patients with breast cancer, rheumatoid arthritis, depression and/or anxiety, stress-related complaints, heart palpitations, and obesity (excluding one disease group at the time).

All statistical analyses were performed using STATA version 15.1 (StataCorp, College Station, TX, USA). *p*-values < 0.05 were considered statistically significant, and all statistical tests were two-sided. 

## 3. Results

### 3.1. Search Results

After removing the duplicates, 3441 records were available for titles and abstract screening. At this stage, 3356 records were excluded as not meeting the PICOS-criteria leaving 85 articles for full-text examination. We further excluded another 75 studies for the following reasons: not relevant outcome (*k* = 20); not relevant study design (*k* = 27); not relevant comparison group (*k* = 1); not relevant intervention (*k* = 14); not relevant study participants (*k* = 1); and incomplete numerical data to retrieve or calculate ES (*k* = 12). The former category (not relevant outcomes) also included studies where pro-inflammatory cytokines were measured in blood cells, not in plasma as in other studies (only one study was excluded due to this reason) since the measure was considered as not comparable to the plasma measures. The latter category (incomplete data) also included studies where authors did not respond to our inquiry on providing additional data. Figure 1 describes a selection process that yielded a final number of 10 studies to be included in the meta-analysis. The complete list of the excluded studies (*k* = 75) with the reasons for exclusion is available on request from the authors.

### 3.2. Characteristics of Included Studies

Table 1 describes the characteristics of studies eligible for inclusion. Out of 10 included studies [36,37,40,41,42,43,50,51,52,53] that in total comprised 607 participants, 9 studies were RCTs [37,40,41,42,43,50,51,52,53], while no randomization was used for allocating patients into intervention and control groups in the study by Lee et al. [36]. Trials were mainly conducted in the USA and Canada [40,42,43,50,51,53], with fewer studies coming from Europe [41,52], New Zealand [37], and South Korea [36]. Furthermore, 8 out of 10 studies were conducted in out-patient clinics [36,37,42,43,50,51,52,53], with community-dwelling individuals being recruited in two other studies [40,41]. Only one study focused on healthy individuals [40], while other studies recruited patients with social anxiety [50], moderate depression and anxiety [52], generalized anxiety disorder (GAD) [51], stress-related complaints [41], breast cancer [36,42], rheumatoid arthritis [37], heart palpitations [43], and post-menopausal BMI > 30 [53]. At least one HRV outcome was assessed in four studies, including LF [41,43,50], HF [41,43,50], LF/HF ratio [36,41,43,50], SDNN and RMSSD [36,41,43], and at least one inflammatory marker was assessed in 6 studies, including CRP [37,40,42,52,53] and IL-6 [40,42,51,53]. Some pro-inflammatory cytokines were only reported in one study each as, for example, IL-8 [52], TNF-α [51], and soluble TNF R2 (as a marker of TNF activity) [42], rendering no data suitable for meta-analysis, and thus the measurements were not included in quantitative synthesis. 

Table 2 provides further details on intervention and comparison groups. Interventions were defined as an 8-week MBSR in 8 original studies [36,37,40,41,43,50,51,52] as a 6-week Mindful Eating and Living (MEAL) intervention [53], and a 6-week Mindful Awareness Practices intervention [42]. All interventions consisted of 2- or 2.5-h weekly meetings. Among 8 studies with MBSR, interventions included weekend retreats of different lengths in 4 studies [37,40,50,51]. Wait-list control was used in one study [40] and treatment as usual (TAU) in 6 studies [36,37,41,42,43,52], while in the remaining 3 trials the comparison groups consisted of a CBT group program [37], stress management education [51], and weight-loss group sessions [53]. The study by Faucher et al. [50] employed two control groups composed of individuals who underwent cognitive behavioral group therapy and healthy volunteers; however, the latter group was not assessed at postintervention and thus was not included in our analysis. 

### 3.3. Assessment of Study Quality

Figure 2 describes quality assessment of each included study. Out of 10 included studies, one study reported using no randomization for allocating the participants into intervention and control group (i.e., nonequivalent control group design) that resulted in assessing the risk of overall selection bias as high. Among the other 9 RCTs, the risk of selection bias due to random sequence generation as well as to allocation concealment was low in more than half of the trials. None of the trials reported blinding of participants and personnel regarding the intervention received, indicating a high risk of performance bias. In contrary, the risk of detection bias due inadequate blinding of outcome assessment as well as the risk of attrition bias due to incomplete outcome data (if missing or excluded from analyses) were mainly low (in 7 out of 10 studies) or unclear (in 3 out of 10 studies). The reporting bias due to selective outcome reporting was assessed as unclear in 9 out of 10 studies and regarded as low in the remaining one trial.

### 3.4. Meta-Analysis of Inflammatory Markers

#### 3.4.1. Interleukin-6

As presented in Figure 3A, by aggregating the results of four studies [40,42,51,53], the pooled estimates for IL-6 yielded a very small effect and did not reach statistical significance (Hedges’ *g* = 0.02; 95% CI −0.29 to 0.34; *p* = 0.88), with an indication of low heterogeneity (Q = 3.70, *p* = 0.296; I^2^ = 18.9%). Publication bias were not evident (Egger’s test *p*-value = 0.49), although the results on publication bias should be interpreted with caution due to small numbers of studies (Figure A1). Influence analysis revealed no individual study, if omitted, to significantly influence the main results (Table A1). In sensitivity analysis, exclusion of study on MBI among breast cancer patients did not alter the pooled estimates obtained in the main analysis and yielded a slightly larger, but still small and non-significant results **(Table A1)**. Likewise, the main results were not altered after excluding study on patients with GAD or a study on postmenopausal women with BMI > 30.

#### 3.4.2. C-Reactive Protein

The aggregated results of five studies on CPR [37,40,42,52,53] revealed small and non-significant effect (Hedges’ *g* = 0.12; 95% CI −0.10 to 0.33; *p*-value = 0.29) and no signs of heterogeneity (Q = 2.24, *p* = 0.69; I^2^ = 0.0%) (Figure 3B). Publication bias were not evident (Egger’s test *p*-value = 0.14) (Figure A1). Influence analysis indicated no studies influencing the main results (Table A1). In a sensitivity analyses, the results remained virtually the same after exclusion of study on breast cancer patients, or rheumatoid arthritis patients, or study on postmenopausal women with BMI > 30, while the pooled estimates become smaller, but remained non-significant after excluding study on patients with moderate depression and anxiety (Table A1). 

### 3.5. Meta-Analysis of Heart Rate Variability Measures

#### 3.5.1. Frequency Domain Measures: LF, HF, and LF/HF Ratio

Pooling the data from three studies [41,43,50] that addressed LF as an outcome of interest yielded inconclusive results with pooled estimates not reaching statistical significance (Hedges’ g = 0.17; 95% CI −0.18 to 0.53; *p*-value = 0.34) and no signs of heterogeneity (Q = 0.73, *p*-value = 0.69; I^2^ = 0.0%) (Figure 4A). Likewise, small and non-significant effect was observed when data on HF from the same studies were pooled (Hedges’ *g* = −0.21; 95% CI −0.88 to 0.45; *p*-value = 0.53), this time with a sign of moderate-to-high heterogeneity (Q = 5.36, *p*-value = 0.07; I^2^ = 62.7%) (Figure 4B). Furthermore, no clear effect of the standardized MBI on LF/HF ratio was noted (Hedges’ g = 0.21; 95% CI −0.26 to 0.67; *p*-value = 0.38), based on the results of four original trials [36,41,43,50], with indication of low heterogeneity (Q = 4.65, *p*-value = 0.19; I^2^ = 0.0%) (Figure 4C). No signs of publication bias were noted for either outcomes (LF: Egger’s test *p*-value = 0.70; HF: Egger’s test *p*-value = 0.86; LF/HF ratio: Egger’s test *p*-value = 0.90), although such a small number of studies for each outcome require caution in interpretation of the results (Figure A2). No individual studies were seen to influence the pooled results for any of the outcomes. (Table A1). Sensitivity analyses revealed no significant alteration for any of the main results that remained non-significant after excluding studies (one at the time) on patients with stress-related complaints, heart palpitations, or social anxiety (Table A1).

#### 3.5.2. Time Domain Measures: SDNN and RMSSD

Three studies addressed SDNN and RMSSD as the outcomes of interest [36,41,43]. Figure 5A presents the results for the analysis for SDNN with medium-size, non-significant effect (Hedges’ *g* = −0.55; 95% CI −1.26 to 0.15; *p*-value = 0.13) and moderate heterogeneity (Q = 4.59, *p*-value = 0.10; I^2^ = 56.5%). No publication bias was noted (Egger’s test *p*-value = 0.19) (Figure A3). The influence analysis, however, indicated a study by Nyklicek et al. [41] to be significantly influencing the pooled estimates. Thus, if excluded, the results become significant, indicating a favour of control (Hedges’ *g* = −0.99; 95% CI −1.71 to −0.27) (Table A1). Sensitivity analyses revealed a significant reduction in SDNN measures if study on community-dwelling individuals with elevated stress level was excluded (Table A1). The results for RMSSD indicated a very small non-significant effect of MBI (Hedges’ *g* = 0.02; (95% CI −0.44 to 0.49, *p*-value = 0.92) with potentially small heterogeneity (Q = 2.50, *p* = 0.29; I^2^ = 20.2%) (Figure 5B). No publication bias was detected (Egger’s test *p*-value = 0.91) (Figure A3) along with no indication of any individual studies to influence the overall ES (Table A1). Sensitivity analyses revealed no alteration for the main results on RMSSD when studies on patients with breast cancer, stress-related complaints, or stress-related complaints were excluded (one at the time) (Table A1).

## 4. Discussion

Our study highlights a substantial scarcity in evidence on the effect of standardized MBIs on inflammatory markers and HRV parameters. The systematic review and meta-analysis ended up with mixed and inconclusive results, assumedly due to the small number of the original studies on each specific outcome and, in particular, to the lack of large, rigorously conducted RCTs. 

### 4.1. Comparison to Existing Literature

A direct comparison to the existing literature is difficult as other reviews mostly assessed the effects of more general mind-body therapies with standardized MBIs being combined together with other interventions of interest. The closest comparison is a recent meta-analysis of workplace-based MBIs where beneficial effects on HRV measures and CRP were found [54]. However, that meta-analysis included studies with longer interventions, as well as interventions with shorter sessions than 2 h (i.e., not falling under the definition of standardized MBIs). Moreover the effect of MBIs on HRV was assessed in only one study on a 12-week intervention, and the only one study with 1-h weekly sessions found significant intervention effects on CRP levels [54]. Both of those studies have been excluded from the present meta-analysis as not fulfilling the inclusion criteria for intervention. Furthermore, a recent meta-analysis on other mind-body interventions (Yoga and Tai-Chi) found small-to-moderate beneficial effects on HRV LF and HF measures [28]. Although MBIs also include movement-based yoga practices, the interventions cannot be considered as identical modalities to standardized MBI and their effects can thus not be compared directly. It might, however, be the case that movement-based practices indeed have more effect on HRV variables than MBIs, in which movement-based practices only comprise a small component. 

Regarding the lack of MBI’s effects on inflammatory markers observed in the present study, our results differ from those found by Black and Slavich [26] in a systematic review on the effects of mindfulness, as well as from findings reported by Morgan et al. [27] in a meta-analysis on the effects of mind-body interventions. The differences could be explained by the broader inclusion criteria used in those reviews. In Black and Slavich’s study [26], positive effects on CRP were found probably due to the inclusion of studies with interventions outside of the standardized MBIs and with a larger variety of sample gathering methods (e.g., flare-induced inflammation). The meta-analysis by Morgan et al. [27], on the other hand, included mind-body interventions in a broader sense (as, for example, Tai Chi and Yoga), which may explain the difference in findings. However, neither one of these meta-analyses found positive effects of MBIs or mind-body interventions on IL-6, thus paralleling the findings of the present study. 

### 4.2. Limitations

The scarcity of studies assessing the effects of the standardized MBIs on pro-inflammatory cytokines and HRV parameters should be considered as major limitations of our review. In particular, the lack of original evidence affected our analysis of HRV parameters with only three original studies reporting data on LF, HF, SDNN, and RMSSD. Scarce evidence regarding the specific physiological measures has been highlighted in other systematic reviews with qualitative and quantitative data syntheses, where the effects of meditation therapies were assessed [26,27]. Furthermore, due to the limited amount of studies, we were unable to assess the moderating effect of age, gender, and type of control groups or to perform a proper subgroup analysis on baseline health status, including stratification by comorbidities and pre-baseline treatment. Although we attempted to address the role of the participants’ health status in the sensitivity analyses, the results are tentative due to a high variety of reported ill-health conditions. The importance of individual demographic and clinical characteristics and the type of controls for the magnitude of physiological effects of MBIs have repeatedly been indicated in other reviews on mind-body exercises [27,28] and work-based mindfulness meditation [54]. Additionally, the original studies varied in quality with performance bias being the most commonly identified weakness. It is, however, worth mentioning that the nature of the intervention makes blinding of participants and personnel not feasible. Finally, our restriction to English-language publications should also be considered as a limitation. Despite no signs of publication bias, it should be acknowledged that a small number of studies on each specific outcome could have affected precision of the tests’ results. 

### 4.3. Clinical Implications and Future Directions

In light of the findings from our study, as well as from the other studies where the effects of non-pharmacological interventions on pro-inflammatory cytokines and HRV have been assessed, the following recommendations for future research can be made: Regarding the effects of movement-based mind-body interventions on HRV measures [54], it may be of interest to assess whether movement-based and stillness-based mind-body practices have differential effects on HRV measures by comparing their effects through an RCT study design. Also, in light of that another meta-analysis, which applied broader inclusion criteria for mindfulness and mind-body interventions than the present study, found positive effects on CRP [27] more well-controlled studies with larger populations, assessing CRP as an outcome may be of interest. 

## 5. Conclusions

The present systematic review and meta-analysis of the effects of the standardized MBIs on inflammatory markers and HRV parameters, when compared to active controls, treatment as usual or wait-list controls, found no significant evidence of effects. The meta-analysis ended up with mixed and inconclusive results, assumedly due to the small number of the original studies on each specific outcome and, in particular, to the lack of high quality studies. In summary, the study findings highlight the necessity of larger, more rigorously conducted RCTs with standardized MBIs being compared to various forms of active controls, also including more long-term follow-ups. 

## Figures and Tables

**Figure 1 jcm-08-01638-f001:**
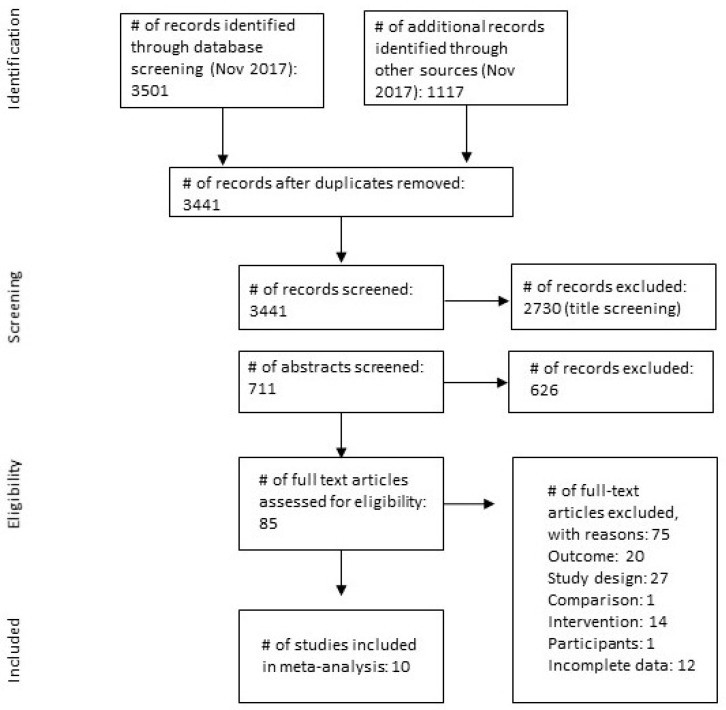
Study selection process. Preferred Reporting Items for Systematic Reviews and Meta-Analyses (PRISMA) flow diagram [30].

**Figure 2 jcm-08-01638-f002:**
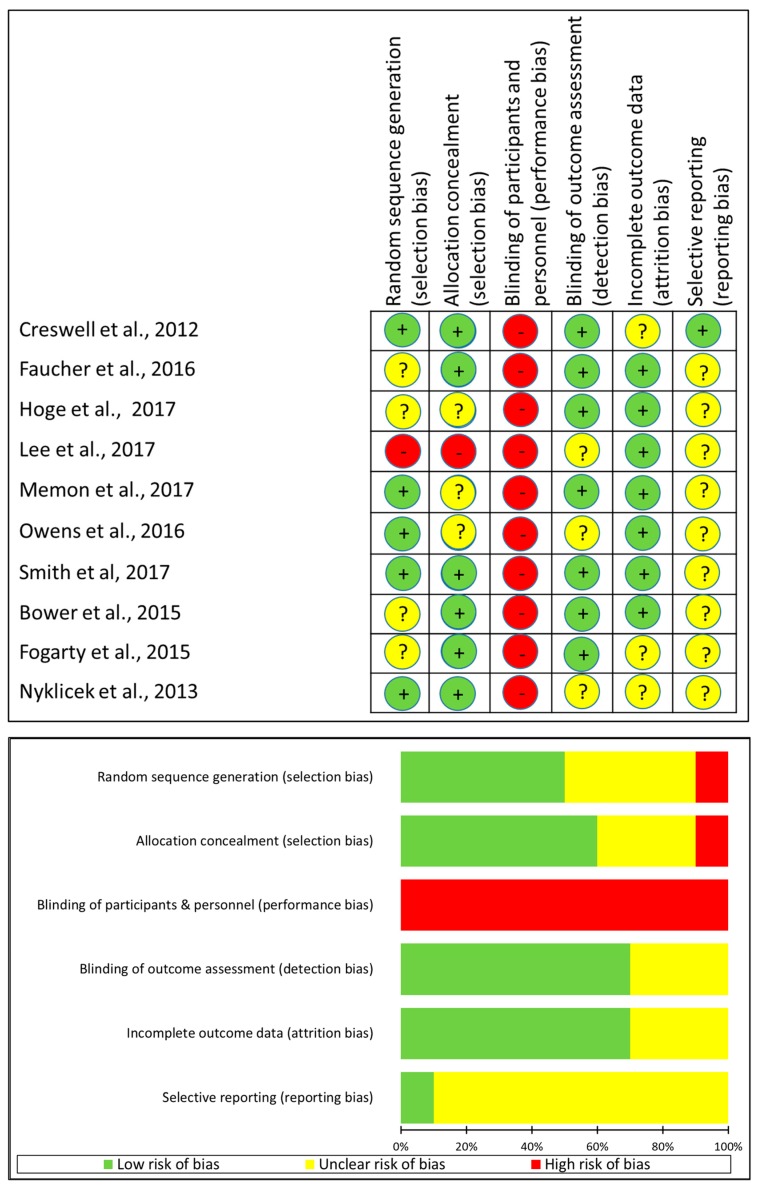
Risk of bias of included studies.

**Figure 3 jcm-08-01638-f003:**
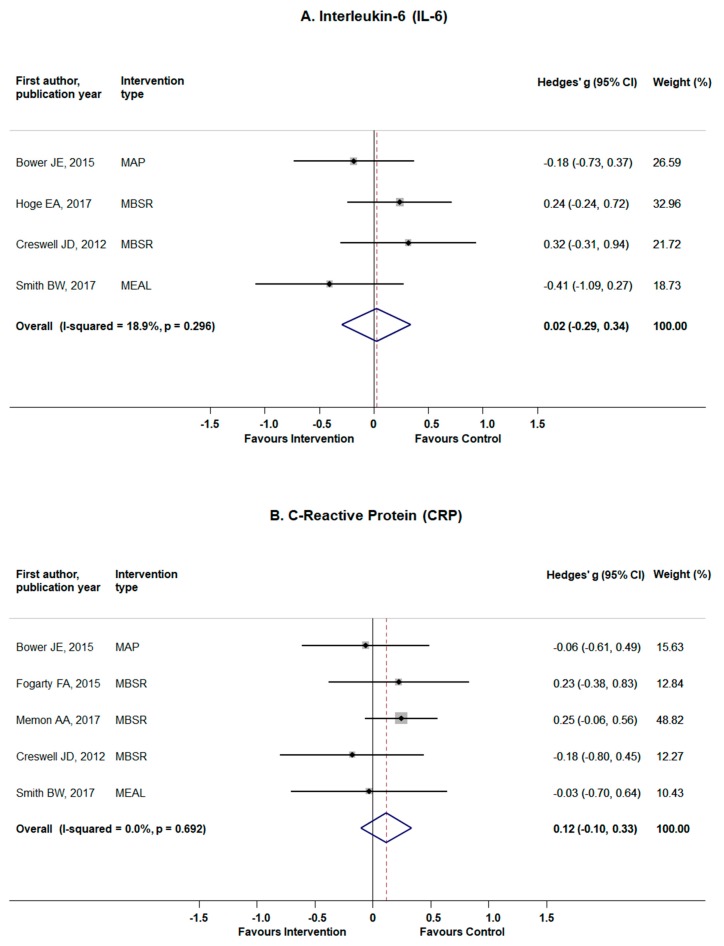
The effects of standardized mindfulness-based interventions on the interleukine-6 (IL-6) (panel **A**) and acute phase reactant protein C-reactive protein (CRP) (panel **B**). Squares indicate standardized difference in means (Hedges’ g) and lines represent 95% confidence intervals (CI); the size of the box represents the weight of each study.

**Figure 4 jcm-08-01638-f004:**
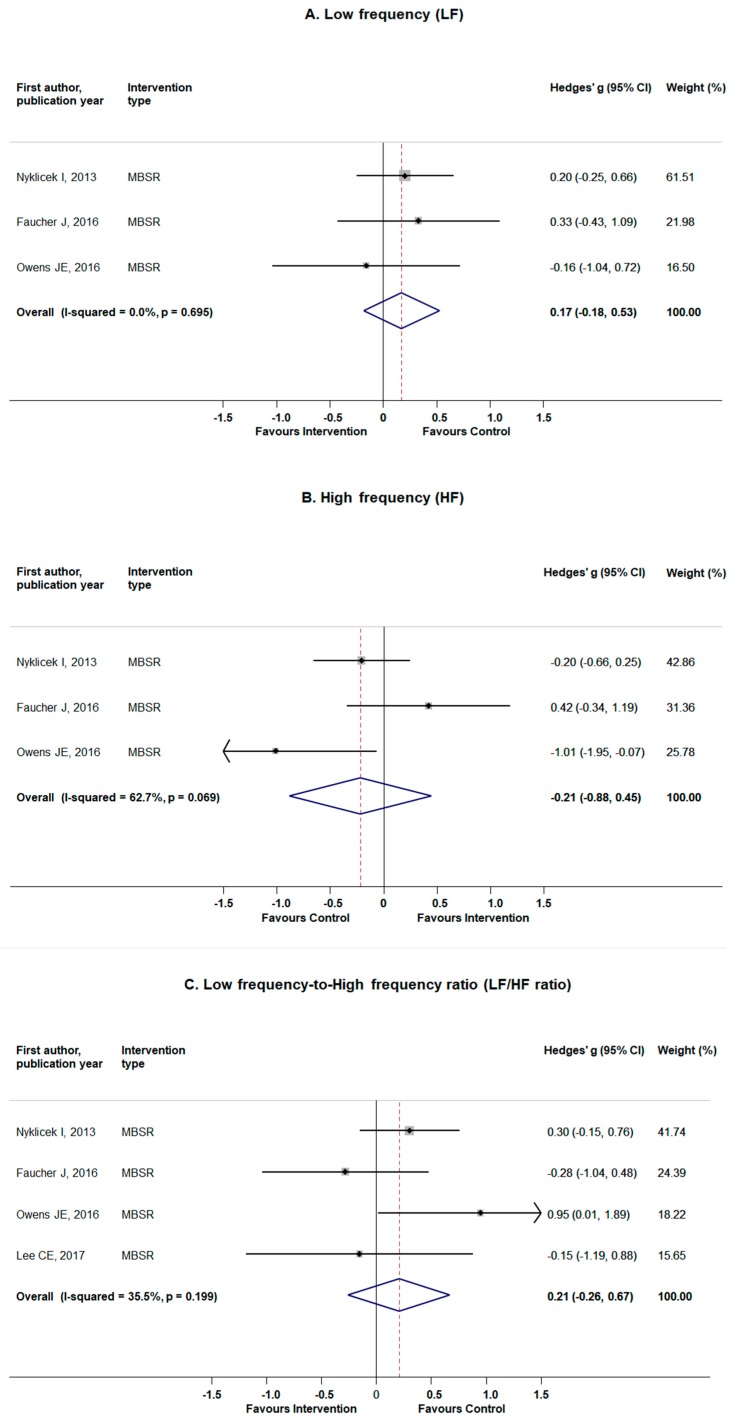
The effects of standardized mindfulness-based interventions on frequency domain heart rate variability (HRV) parameters: low frequency (LF) (panel **A**), high frequency (HF) (panel **B**), and LF/HF ratio (panel **C**). Squares indicate standardized difference in means (Hedges’ g) and lines represent 95% confidence intervals (CI); the size of the box represents the weight of each study.

**Figure 5 jcm-08-01638-f005:**
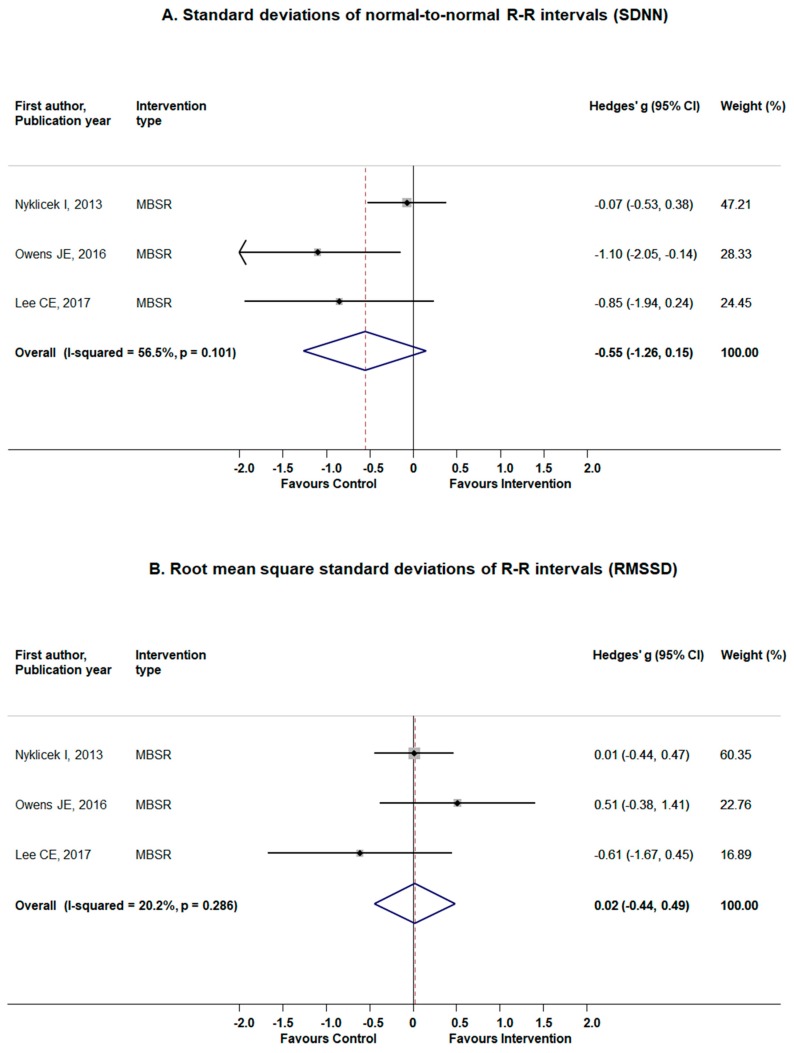
The effects of standardized mindfulness-based interventions on time domain heart rate variability (HRV) parameters: standard deviations of normal-to-normal R-R intervals (SDNN) (panel **A**) and root mean square standard deviations of R-R intervals (RMSSD) (panel **B**)**.** Squares indicate standardized difference in means (Hedges’ g) and lines represent 95% confidence intervals (CI); the size of the box represents the weight of each study.

**Table 1 jcm-08-01638-t001:** Study summary.

First Author (Year)	Country	Setting	Inclusion Criteria	Design	Outcome	Main Findings
Bower et al. (2015)	USA	Out-patient	Patients with diagnosis of stage 0, I, II, or III breast cancer at or before age 50 years; and who had completed local and/or adjuvant cancer therapy (except hormone therapy)	RCT comparing MAP (mindful awareness practices) with TAU	IL-6	There were no significant intervention effects for IL-6 (*p* > 0.20 for both).
Creswell et al. (2012)	USA	Non-clinical	Healthy older adults who indicated an interest in learning mindfulness meditation techniques, English-speaking, not currently practicing any mind–body therapies, non-smokers, mentally and physically healthy for the last three months, and not currently taking medications that affect immune, cardiovascular, endocrine, or psychiatric functioning	RCT comparing MBSR with a wait-list control group	IL-6 and CRP	There was a trend for MBSR to reduce CPR (treatment condition - time interaction): (F(1,33) = 3.39, *p* = 0.075).
Faucher et al. (2016)	Canada	Out-patient	Outpatients with social anxiety disorder, according to DMS-IV criteria, and score > 50 on Liebowitz Social Anxiety Scale, and score > 4 on Clinical Global Impression of Illness subscale, medication-free.	RCT comparing MBSR with a CBT group program	HRV (LF, HF and LF/HF)	No physiological differences were found as a function of treatment
Fogarty et al. (2015)	New Zealand	Out-patient	Patients with reumathoid arthritis, according to the 1987 American College of Rheumatology classification criteria and without any prior meditation experience	RCT comparing MBSR with TAU	CRP	There were no significant group-time effects on CRP levels
Hoge et al. (2017)	USA	Out-patient	Individuals age 18 or older were eligible if they: (a) met DSM-IV criteria forcurrent primary GAD and designated GAD as the primary problem, and (b) scored 20 or above on the Hamilton Anxiety scale (HAM-A).	RCT comparing MBSR with additional metta (loving-kindness meditation) already in the first class, compared to an active control consisting of Stress Management Education (attention control)	IL-6	The MBSR group had a greater reduction in inflammatory cytokines IL-6 AUC concentrations compared to controls
Lee et al. (2017)	South Korea	Out-patient	Patients diagnosed with metastatic breast cancer who were currently undergoing anti cancerous treatment in an outpatient clinic, were 20 years of age or older, and were able to read and write in Korean	Non-randomized controlled trial with non-equivalent control group comparing MBSR with TAU	HRV (SDNN, RMSSD, LF/HF)	For HRV, although there was no significant difference between the groups for SDNN, RMSS, total power, and LF/HF, improved tendencies were observed in the MBSR group for SDNN from 24.81 to 53.93 (*p* = 0.051)
Memon et al. (2017)	Sweden	Out-patient	Patients with mild to moderate depression and anxiety, aged between 20 and 64 years, were fluent in Swedish and had a score of ≥10 on the PHQ-9, ≥7 on the HADS-D or HADS-A or a score on the MADRS between 13 and 34.	RCT comparing MBSR with TAU (including CBT and pharmacological treatment for some patients)	IL-6 and hsCRP	Levels of inflammatory markers analyzed in this study, were not significantly associated with treatment response on any scale.
Nyklicek et al. (2013)	The Netherlands	Non-clinical	People having stress-related complaints, potential participants were eligible if they answered with "regularly" or "often" to the question “how often would you say you feel distressed?”	RCT comparing MBSR with TAU	HRV (SDNN, RMSSD, LF/HF, HF and LF)	No effects were obtained on HRV measures.
Owens et al. (2016)	USA	Out-patient	Patients reporting heart palpitations of at least two months duration, willingness to attend MBSR classes and comply with the data collection protocol.	RCT comparing MBSR with TAU	HRV (SDNN, RMSSD, LF/HF, HF and LF)	There were no significant differences between the MBSR and Control groups on any of the HRV measures at baseline, 8 weeks, or 12 weeks. An association was found between HRV balance (as measured by the Ln LF/HF ratio) and improvement in palpitations in the MBSR group (r = 0.8, *p* < 0.001)
Smith et al. (2017)	USA	Out-patient	Women aged 50–70 years with post-menopausal status, a BMI of more than 30, ability to participate in the study for 1 year, fluency in English, and ability to walk at least 10 min without stopping.	RCT comparing MEAL (Mindful eating and living) with a group session with same schedule as the intervention	IL-6, CRP	The reductions in IL-6 and CRP were significantly greater for the MEAL as compared with the control group.

Table legend: Abbreviations: Randomised Controlled Trial (RCT); Mindfulness Based Stress Reduction (MBSR); Cognitive Behavioural Therapy (CBT); Treatment as usual (TAU); interleukin 6 (IL-6); C-reactive protein (CRP); high sensitivity (hsCRP); Heart Rate Variability (HRV); Low frequency (LF); High frequency (HF); standard deviations of normal-to-normal R-R intervals (SDNN); root mean square standard deviations of R-R intervals (RMSSD).

**Table 2 jcm-08-01638-t002:** Summary of participant and intervention characteristics.

	Participant Characteristics			Intervention and Control Condition Name and Duration
First Author (Year)	Mean Age and (Range) in Years	Female (%)	*N* Total	
Bower et al. (2015)	I: 46.1 (28.4–60.0) C: 47.7 (31.1–59.6)	I: 100%; C: 100%	65	A 6-week Mindful Awareness Practices intervention consisting of 2-h weekly meetings. Comparison: TAU
Creswell et al. (2012)	I: 64.35 (N/A) C: 65.16 *(N/A)*	I: 85%; C: 75%	40	An 8-week Mindfulness Based Stress Reduction intervention consisting of 2-h weekly meetings and a 7-h weekend day retreat. Comparison: wait-list control
Faucher et al. (2016)	I: 36.64 (N/A) C: 39.31 (N/A)	I: 35.7% C: 38.5%	38	An 8-week Mindfulness Based Stress Reduction intervention consisting of 2.5 h weekly meetings and a 7.5 h weekend day retreat. Comparison: a 12-week Cognitive Behavioural Group Therapy intervention consisting of 2.5 h weekly meetings (included psychoeducation, exposure, cognitive restructuring and homework assignments).
Fogarty et al. (2015)	I: 52 (N/A) C: 55 (N/A)	I: 91%; C: 86%	51	An 8-week Mindfulness Based Stress Reduction intervention consisting of 2-h weekly meetings and a full day weekend retreat. Comparison: TAU
Hoge et al. (2017)	I: 40 (N/A) C: 38 (N/A)	I: 43C: 50	70	An 8-week Mindfulness Based Stress Reduction consisting of 2-h weekly meetings and a 4-h weekend retreat, including an additional loving-kindness practice introduced already at the first session. Comparison: Stress Management Education lectures on overall health and wellness such as diet, exercise, sleep, and time management.
Lee et al. (2017)	I: 52 (33–64) C: 57 (37–67)	I: 100%; C: 100%	32	An 8-week Mindfulness Based Stress Reduction intervention consisting of 2 h weekly meetings. Comparison: TAU
Memon et al. (2017)	I: 42 (N/A) C: 41 (N/A)	I: 83%; C: 92%	166	An 8-week Mindfulness Based Stress Reduction intervention consisting of 2 h weekly meetings. Comparison: TAU (including CBT and pharmacological treatment for some patients)
Nyklicek et al. (2013)	I: 47.4 (N/A) C: 44.9 (N/A)	I: 65% C: 76%	85	An 8-week Mindfulness Based Stress Reduction intervention consisting of 2.5-h weekly meetings. Comparison: TAU
Owens et al. (2016)	I: N/A (N/A) C: N/A (N/A)	I: N/A; C: N/A%	20	An 8-week Mindfulness Based Stress Reduction intervention consisting of 2.5-h weekly meetings. Comparison: TAU
Smith et al. (2017)	I: 58.56 (N/A) 58.56 (N/A)	I: 100% C: 100%	40	A 6-week Mindful Eating and Living (MEAL) intervention consisting of 2-h weekly meetings. Comparison: A control program created to match the intervention, and consisting of weight loss group sessions conducted according to the same schedule as the MEAL group.

Table legend: I = intervention; C = control; N/A = data not available; treatment as usual (TAU); Cognitive Behavioural Therapy (CBT).

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
