# Peer review of "A Systematic Review and Meta-Analysis of the Impact of Mindfulness Based Interventions on Heart Rate Variability and Inflammatory Markers"

_jcm, 2019, doi:10.3390/jcm8101638_

Round 1
Reviewer 1 Report
I am very grateful for the opportunity to have reviewed this article and I applaud the enormous honesty with which the authors have proceeded.
Clearly, the results they have found will not satisfy MBIs followers very much. It is likely that they have presented this paper to other journals (it is just guessing) because the work is very polished and well written (except for a couple of things that I will add below).
Nevertheless, I recommend its publication FOR A SIMPLE AND POWERFUL SCIENTIFIC REASON: Significant results are just as important as non-significant results. And they have correctly pointed out the reasons why MBIs (at least HRV and inflammatory markers) are still immature.
An important high IF journal like Mindfulness specifically points out for MBIs papers: what is the level of MBI experiences have the instructor or therapist, how much time last his/her/their MBI learning, where did they were taught and how much time MBI instructors spent in the intervention. These suggestions of Mindfulness are based on this kind of paper. Meta-analysis about MBIs is still varying in strength and direction.
To my knowledge, MBI requires of people well experienced and instructed. Many papers have no provided this remarkable information. o
Minor revisions and recommendations
In order to give more credibility to their conclusions, I missed that they have not included in a table all the keywords and boolean procedure that they used in their databases search. In this way, any other devout author of the mindfulness can arrive at the same result or not.
Although the references follow the recommendations of the journal, I would recommend that they include an asterisk for the 10 selected references for systematic review and meta-analysis. For example, 42* Bower et al........
Bower et al's study do not show the mean age of the control condition group. Surely, it was an error
Best wishes to authors
Author Response
REVIEWER #1
AUTHORS’ RESPONSE: We wish to thank the reviewer for the encouraging comments on our paper, we provide our responses below in bold.
Minor revisions and recommendations
In order to give more credibility to their conclusions, I missed that they have not included in a table all the keywords and Boolean procedure that they used in their databases search. In this way, any other devout author of the mindfulness can arrive at the same result or not.
AUTHORS’ RESPONSE: Thank you for the comment. We fully support the idea of transparency in reporting the search procedure that ensures reproducibility of our results. In our paper such procedure was reported in details in the Supplementary Material 1 (submitted together with the manuscript as a separate file). In the manuscript, under the subheading “2.2. Search strategy”, the corresponding information was added (lines 124-124 in the Track-changed version).
Below is a quote from the revised manuscript:
“Free text or index terms (e.g. Medical Subject Headings) were searched for in titles, abstracts and key words (see Supplementary Material 1 for details on searched terms and the use of Boolean operators). The use of key words was adapted to each electronic database (see Supplementary Material 1 for details)”.
Added to abovementioned, we would like to report that we found a typo in the Supplementary Material 1 where we erroneously indicated the date of search for some datasets as “2016-11-09” instead of the correct date “2017-11-09”. We have now corrected the typo and uploaded the revised Supplementary Material 1 file while re-submitting the manuscript.
Although the references follow the recommendations of the journal, I would recommend that they include an asterisk for the 10 selected references for systematic review and meta-analysis. For example, 42* Bower et al........
AUTHORS’ RESPONSE: We are grateful for the suggestion on marking the references selected for the analysis. Following the comment, we added an asterisk to the corresponding references. To clarify the use of asterisk, we have provided a footnote right below the reference list (line 577 in the Track-changed version).
Below is a quote from the revised manuscript:
“* References with asterisk are included in meta-analysis”.
Bower et al's study do not show the mean age of the control condition group. Surely, it was an error
AUTHORS’ RESPONSE: We are grateful to the reviewer for pointing this error out. We have added the missing mean age for the control group in the study by Bower to Table 2.
Below is a quote from the revised manuscript:
“C: 47.7 (31.1-59.6)”.
Reviewer 2 Report
The manuscript is well done, precise, punctual and exhaustive.
The only comment I could make concerns the abstract that I would
reformulate so that it differs, at least in the form, from the writing
of the article.
Author Response
REVIEWER #2
AUTHORS’ RESPONSE: We thank the reviewer for positive comments and provide our response in bold letters.
The only comment I could make concerns the abstract that I would reformulate so that it differs, at least in the form, from the writing of the article.
AUTHORS’ RESPONSE: We are grateful for the comment and have revised the text in the Abstract in a way that it does not repeat verbatim the text of the manuscript. The changes are tracked in the manuscript (line 18-21 and 30-34 in the Track-changed version).